# Structure–Activity Relationship Studies of 9-Alkylamino-1,2,3,4-tetrahydroacridines against *Leishmania* (*Leishmania*) *infantum* Promastigotes

**DOI:** 10.3390/pharmaceutics15020669

**Published:** 2023-02-16

**Authors:** Carlos F. M. Silva, Teresa Leão, Filipa Dias, Ana M. Tomás, Diana C. G. A. Pinto, Eduardo F. T. Oliveira, Ana Oliveira, Pedro A. Fernandes, Artur M. S. Silva

**Affiliations:** 1LAQV-REQUIMTE, Department of Chemistry, University of Aveiro, 3810-193 Aveiro, Portugal; 2i3S, Instituto de Investigação e Inovação em Saúde, 4200-135 Porto, Portugal; 3ICBAS, School of Medicine and Biomedical Sciences, Universidade do Porto, 4050-313 Porto, Portugal; 4UCIBIO, REQUIMTE, Departamento de Química e Bioquímica, Faculdade de Ciências, Universidade do Porto, 4169-007 Porto, Portugal

**Keywords:** *Leishmania*, virtual screening, AdoMet DC, microwave-assisted synthesis, 1,2,3,4-tetrahydroacridine, quinoline, promastigotes, macrophages, molecular docking

## Abstract

Leishmaniasis is one of the most neglected diseases in modern times, mainly affecting people from developing countries of the tropics, subtropics and the Mediterranean basin, with approximately 350 million people considered at risk of developing this disease. The incidence of human leishmaniasis has increased over the past decades due to failing prevention and therapeutic measures—there are no vaccines and chemotherapy, which is problematic. Acridine derivatives constitute an interesting group of nitrogen-containing heterocyclic compounds associated with numerous bioactivities, with emphasis to their antileishmanial potential. The present work builds on computational studies focusing on a specific enzyme of the parasite, *S*-adenosylmethionine decarboxylase (AdoMet DC), with several 1,2,3,4-tetrahydro-acridines emerging as potential inhibitors, evidencing this scaffold as a promising building block for novel antileishmanial pharmaceuticals. Thus, several 1,2,3,4-tetrahydroacridine derivatives have been synthesized, their activity against *Leishmania (Leishmania) infantum* promastigotes evaluated and a structure–activity relationship (SAR) study was developed based on the results obtained. Even though the majority of the 1,2,3,4-tetrahydroacridines evaluated presented high levels of toxicity, the structural information gathered in this work allowed its application with another scaffold (quinoline), leading to the obtention of *N*^1^,*N*^12^-bis(7-chloroquinolin-4-yl)dodecane-1,12-diamine (**12**) as a promising novel antileishmanial agent (IC_50_ = 0.60 ± 0.11 μM, EC_50_ = 11.69 ± 3.96 μM and TI = 19.48).

## 1. Introduction

By the beginning of the twenty-first century, the World Health Organization (WHO) recognized 20 diseases as being neglected tropical diseases, including leishmaniasis [1,2]. Leishmaniasis mainly affects people from developing countries from the tropics, subtropics and the Mediterranean basin, with an estimated world prevalence of 12 million cases covering all forms of leishmaniasis [3,4]. Moreover, these numbers tend to increase, since 1.5–2 million new cases of cutaneous leishmaniasis and 500,000 cases of visceral leishmaniasis appear each year, with a mortality rate of approximately 50,000 deaths annually. The etiologic agent of the disease is a protozoan parasite, *Leishmania*, that exists in two distinct forms, an extracellular promastigote within the sand-fly and an obligate intracellular amastigote within cells of the mammalian host monocyte-macrophage lineage [5]. Considering the present scenario, with the currently used medicines presenting limitations in both toxicity and affordability, it becomes mandatory to develop new antileishmanial agents. Thus, the search for novel antileishmanial drugs should be focused on finding less toxic and easily available (within the reach of people below the poverty line, who are most afflicted by the disease) products, with plant-derived compounds emerging as promising starting points for the process of drug discovery and development [6,7]. *Leishmania* appears to be a highly complex parasite that contemplates a considerable variety of metabolic pathways, with several enzymes suitable to be potential targets for molecular docking guided small molecule inhibition [8]. One of these, *S*-adenosylmethionine decarboxylase (AdoMet DC, EC 4.1.1.50), was selected as one of the most promising drug targets given its crucial role in the polyamine biosynthesis of the parasite by being responsible for converting *S*-adenosylmethionine into *S*-adenosylmethylthiopropylamine. In turn, the latter was then used as a precursor for the synthesis of both spermidine and spermine, which corroborates the influence of AdoMet DC in polyamine biosynthesis. Additionally, by influencing the synthesis of spermidine, this enzyme indirectly impacts thiol metabolism by also affecting the synthesis of trypanothione (TS_2_). AdoMet DC is regulated by a peculiar mechanism, with the enzymatically active complex present as a heterodimer consisting of a functional subunit and a catalytically inactive paralog, a unique characteristic of the kinetoplastids that might facilitate the development of selective treatments without affecting the mammal host [9,10]. Moreover, the inhibition of this enzyme has already been associated with the cure of animal leishmaniasis, so it might be a promising target for treating human leishmaniasis [11].

Acridine derivatives constitute an interesting group of nitrogen-containing heterocyclic compounds, known for their wide range of pharmaceutical properties [12,13]. This class of compounds has caught the attention of the scientific community, mainly due to its unique physical and chemical properties, which enable several derivatives to be associated with numerous bioactivities, including antiparasitic, antimalarial and antiviral activities, amongst many others. Even though the full potential of these types of derivatives as antileishmanial agents remains to be determined, there are already some reports of acridine derivatives with interesting antileishmanial effects, leaving some possibilities to develop 1,2,3,4-tetrahydroacridines as novel potential antileishmanial agents [14]. The aim of this study is to find potential hit compounds against *L. (L.) infantum*, based on the virtual screening of a considerably large chemical library of natural compounds against AdoMet DC, with particular focus on 1,2,3,4-tetrahydroacridines.

## 2. Materials and Methods

### 2.1. General

Commercially available reagents were used without any previous purification and were obtained from Sigma Aldrich (San Louis, MO, USA). The solvents used during both reactions and purification procedures were analytically pure or were, if necessary, dried using the appropriate molecular sieves of 3Å. The MW-assisted syntheses were performed in dedicated CEM Discovery SP monomode microwave equipment. The progress of the chemical reactions was continuously evaluated through thin layer chromatography (TLC) in aluminum sheets coated with silica gel 60 G/UV254 from Merck (Burlington, VT, USA). The purification procedures through preparative thin layer chromatography were performed using glass plates (20 cm × 20 cm × 0.5 cm) previously coated with a layer of silica gel 60 G/UV254 from Macherey-Nagel. The purifications through column chromatography were performed with silica gel 60 (0.060–0.200 mm) from Acros Organics.

### 2.2. Instrumentalization

The NMR spectra of our compounds were mainly obtained in Bruker Avance 300 equipment, operating at 300.13 MHz for ^1^H and 75.47 MHz for ^13^C. The exceptions were obtained in a Bruker Avance 500 (Billerica, MA, USA), operating at 500.13 MHz for ^1^H and 125.77 MHz for ^13^C. All these analyses were performed with tetramethylsilane (TMS) as internal standard, with the chemical shifts of each signal reported in parts per million (ppm) and the coupling constants (*J*) in Hertz (Hz). The melting points of the synthesized compounds were determined using Buchi Melting Point B-540 equipment (Flawil, Switzerland) and, when available, compared with the literature. Positive-ion ESI mass spectra were acquired with a QTOF_2_ instrument [dilution of 1 μL] of the sample in chloroform solution (ca. 10–5 mL) in 200 μL of 0.1% trifluoroacetic acid/methanol solution. Nitrogen was used as the nebulizer gas and argon as the collision gas. The needle voltage was set at 3000 V, with the ion source at 80 °C and the desolvation temperature at 150 °C. The cone voltage was 35 V. Other low-and high-resolution mass spectra (EI, 70 eV) were measured with VG Autospec Q and M spectrometers.

### 2.3. Computational Studies

#### 2.3.1. Virtual Screening

Our hit discovery strategy comprised two approaches: a ligand-based virtual screening to restrict the chemical space of interest, followed by a structure-based virtual screening that ranked the hits according to their ability to bind to the enzyme *S*-adenosylmethionine decarboxylase (EC 4.1.1.50).

The first step of the structural studies was to identify in the Chembl database active molecules already tested in this particular target. Results retrieved 50–60 potential active compounds. Then, we searched for related compounds, with similar substructures to the above mentioned as potentially active compounds using the Chembl and ZINC databases. From this search, 99,138 compounds were found, and their 3D models were consecutively generated.

Second, a structure-based virtual screening of this large dataset was performed to identify enzyme hits. Here, we started by setting up the protein for the vs. studies. Furthermore, a crystallographic structure of a complex enzyme-inhibitor belonging to *T. brucei* (5TVM) was also available in the protein data bank (PDB) with the protonation states already attributed. These data were then used for the elaboration of a pharmacological profile which allowed the determination of some docking parameters, such as the identification of interaction regions important to the biological activity. It is important to mention that, since there was no structure available of a *Leishmania* AdoMet DC crystallographic structure complexed with an inhibitor, a *T. brucei* alternative was used due to the similarity between the two species’ enzyme (61% sequence identity, 74% sequence similarity).

Then, to choose a scoring function and optimize the protocol for this particular study, the top 32 active compounds found in the CHEMBL database were used as input to generate decoys. Fifty decoys were generated for each of them using the DUD-E server, comprising 1632 compounds in total. We then tested several scoring functions implemented in different molecular docking software and concluded that two combined scoring functions implemented in Autodock VINA and GOLD software (Astex Statistical Potential) should be used to achieve the best results. Based on this, a protocol for successful hit identification was set, and a structure-based virtual screening was run using the 99K compounds library. Finally, we clustered the compounds based on their chemical diversity.

#### 2.3.2. Molecular Docking

To exploit in detail the structurally similar derivatives synthesized in this project, we carried out molecular docking calculations of the most promising compounds in the active site of the AdoMet DC from *L. (L.) infantum*. First, the accuracy of our docking protocol was validated by extracting the co-crystalized inhibitor (CGP 40215) of the AdoMet DC from *T. brucei* TREU927 (PDB ID: 5TVF) and redocking the compound on the enzyme active site. Then, the AlphaFold2 model of the (AF-Q9NGA0-F1) was used as input for the protein, while the compounds were manually designed using the Marvin Sketch software and the enzyme’s ionization states were predicted using the propKA 3.1 software. Then, the ligands were submitted to molecular docking calculations on the AdoMet DC enzymes of *L. (L.) infantum* and *T. brucei* using the GOLD software. In the calculations, we defined a 15 Å radius cavity, centered on the position occupied by the co-crystalized ligand in the 5TVF structure, and allowed full flexibility of the ligands using the genetic algorithm (GA) searching algorithm. As output, we obtained 50 docking solutions for each and the top 10 were visually inspected using the PyMol 2.5.4 software.

### 2.4. Synthesis

#### 2.4.1. General Procedure for the Synthesis of 9-Chloroacridines (**5**)

[Conventional] These syntheses were performed following the procedure described by Hu et al. [15]. To a round bottom flask, the appropriate 2-aminobenzoic acid (0.100 g, **3.a** [0.7292 mmol], **3.b** [0.5828 mmol]) and cyclohexanone (**4**, 0.7291 mmol for **3.a** and 0.5828 mmol for **3.b**, 1 equiv.), an excess of POCl_3_ (2.0 mL) were added, with the reaction left stirring at reflux for 2 h. After, the reaction mixture was poured onto ice, neutralized with Na_2_CO_3_ to pH 7, and a liquid–liquid extraction was performed using ethyl acetate (3 *×* 20 mL). Finally, the solvent was evaporated to dryness and the crude purified through preparative thin layer chromatography, using dichloromethane as eluent.

[MW-assisted] The appropriate 2-aminobenzoic acid (**3.a** [0.7292 mmol], **3.b** [0.5828 mmol]) and cyclohexanone (**4**, 0.7292 mmol for **3.a** and 0.5828 mmol for **3.b**, 1 equiv.) were added to a MW vessel (10 mL). Then, an excess of POCl_3_ (2.0 mL) was further added to the reaction mixture, with the MW vessel (10 mL) being placed into the monomode MW (potency = 150 W) at 150 ˚C for 30 min. After its conclusion, the workup and purification followed the procedure described for the conventional method.

#### 2.4.2. General Procedure for the Synthesis of 9-Alkylamino-1,2,3,4-tetrahydroacridines (**6.a–n**)

These syntheses were performed following the procedure described by Hu et al. [15]. The appropriate 1,2,3,4-tetrahydroacridine (**5.a** [0.4593 mmol], **5.b** [0.3966 mmol]), the selected alkyldiamine (1.2 equiv.), 10% (*w*/*w*) of NaI and phenol (2.297 mmol for **5.a** and 1.983 mmol for **5.b**, 5 equiv.) were transferred into a MW vessel (10 mL). Then, the MW vessel (10 mL) was placed into the monomode MW (potency = 150 W) at 150 °C for 30 min. After its conclusion, the reaction mixture was cooled to room temperature, the residue dissolved in ethyl acetate and a liquid–liquid extraction performed, using an aqueous solution of KOH (10%, 20.0 mL). The organic phase was then isolated, the solvent evaporated to dryness and the crude purified through preparative thin layer chromatography using a (9:1) mixture of ethyl acetate/triethylamine as eluent.

#### 2.4.3. General Procedure for the Synthesis of *N*-Acetylated-9-alkylamino-1,2,3,4-tetrahydroacridines (**7.a–n**)

The freshly synthesized 9-alkylamino-1,2,3,4-tetrahydroacridine (**6**, 0.500 mmol) was dissolved with methanol (3.0–5.0 mL) in a MW vessel (10 mL) and an excess of acetic anhydride (1.0 mL) and a few drops of concentrated H_2_SO_4_ (96%) were added. The MW vessel (10 mL) was then placed in the monomode MW (potency = 50 W) at 100 °C for 10 min. After, the solvent was evaporated to dryness, the crude dissolved in ethyl acetate (10.0 mL) and a liquid–liquid extraction was consequently performed using water (10.0 mL). Finally, the organic phase was isolated, the solvent evaporated to dryness and the product recovered in a quantitative yield, without the need for further purification.

#### 2.4.4. General Procedure for the Synthesis of *N*-Formylated-9-alkylamino-1,2,3,4-tetrahydroacridine (**8.a–f**)

The freshly synthesized 9-alkylamino-1,2,3,4-tetrahydroacridine (**6.e–g**,**l–n**, 0.500 mmol) was dissolved in DMF (3.0 mL) in a MW vessel (10 mL), and triethylamine (0.210 mL, 1.500 mmol, 3 equiv.) was added to the reaction mixture. The MW vessel (10 mL) was then placed in the monomode MW (potency = 150 W) at 120 °C for 30 min. After its conclusion, the reaction mixture was cooled to room temperature and a liquid–liquid extraction was consequently performed using ethyl acetate (2 × 10.0 mL) and water (10.0 mL). Finally, the organic phase was isolated, the solvent evaporated to dryness and the crude purified through preparative thin layer chromatography using a (9:1) of mixture ethyl acetate/triethylamine as eluent.

#### 2.4.5. General Procedure for the Synthesis of Compounds **9.a–d**

To a round bottom flask, 2,5-dichloro-1,4-benzoquinone (0.5650 mmol) was transferred and dissolved in ethanol (5.0 mL), followed by the slow addition of Na_2_CO_3_ (1.130 mmol, 2 equiv.). Lastly, an excess of the desired 9-aminated tetrahydroacridine (**6.c** or **6.i**, 1.413 mmol, 2.5 equiv.) was also added to the reaction mixture and left to react for 3–6 h at room temperature. The evolution of this reaction was then evaluated through thin layer chromatography and, when the starting materials were completely consumed, the solvent was evaporated and a liquid–liquid extraction was performed with dichloromethane (3 × 20 mL) and a saturated aqueous solution of NaHCO_3_. Finally, the organic phase was isolated, the solvent evaporated to dryness and the resulting crude purified through a preparative thin layer chromatography using ethyl acetate as eluent.

#### 2.4.6. General Procedure for the Synthesis of *N*^1^,*N*^12^-bis(7-chloroquinolin-4-yl)dodecane-1,12-diamine (**12**)

1,12-dodecanodiamine (0.4991 mmol), 4,7-dichloroquinoline (**11**, 1.248 mmol, 2.5 equiv.), 10% (*w*/*w*) of NaI and phenol (2.496 mmol, 5 equiv.) were transferred into a MW vessel (10 mL). Then, the MW vessel (10 mL) was placed into the monomode MW (potency = 150 W) at 150 °C for 30 min. After its conclusion, the reaction mixture was cooled to room temperature, the residue dissolved in ethyl acetate and a liquid–liquid extraction performed, using an aqueous solution of KOH (10%, 20.0 mL). The organic phase was then isolated, the solvent evaporated to dryness and the crude purified through preparative thin layer chromatography using a (9:1) mixture of ethyl acetate/triethylamine as eluent.

### 2.5. Biological Evaluation

#### 2.5.1. Promastigote Stage of the Parasite

The *Leishmania (L.) infantum* promastigotes (strain MHOM/MA/67/ITMAP-263) used in this project were regularly passaged through NMRI mice (infected intravenously with 1 *×* 10^7^ stationary promastigotes) and recovered from the spleen one week after infection. Promastigotes were routinely cultivated at 25 °C in complete RPMI medium, i.e., RPMI 1640 GlutaMAXTM-I medium supplemented with 10% (*v*/*v*) heat-inactivated fetal bovine serum (FBSi), 50 U/mL penicillin, 50 μg/mL streptomycin (all from Gibco) and 20 mM HEPES pH 7.4. In the plate assays carried out to evaluate the anti-promastigote activity, 100 µL of early log stage promastigotes (final concentration of 1 *×* 10^6^ cells/mL) were seeded in complete Schneider’s medium, i.e., Schneider’s medium (Sigma) supplemented with 10% (*v*/*v*) FBSi, 100 U/mL penicillin, 100 μg/mL streptomycin, 5 mM HEPES (pH 7.4) and 5 mg/mL Phenol-red (Sigma). Parasite density was evaluated by cell counting in a Neubauer chamber.

The compound’s stock solutions were prepared in DMSO and then diluted to 200 μM in Schneider’s medium, guaranteeing a percentage of DMSO below 5%. After performing two-fold serial dilutions of the drug, plates were incubated for 24 h at 25 °C. After this period, 20 μL of 0.32 mg/mL resazurin (Sigma) was added to each well, followed by 4 h of incubation. Cell viability was then quantified on a plate reader (Synergy Mx microplate reader, BioTek Instruments, Winooski, VT, USA) using the following settings: excitation, 540/25 nm; emission: 620/40 nm. Data were converted to ‘% Survival’ by first subtracting experimental values from the average of media-only control, and then expressing this value as a percentage proportion of the cell-only control. The half-maximal inhibitory concentrations (IC_50_s) were derived from nonlinear regression analysis, using the “log(inhibitor) vs. response (three parameters)” function in GraphPad Prism 8 (GraphPad Software, Inc., La Jolla, CA, USA).

#### 2.5.2. Toxicity against Mammalian Macrophages

The bone marrow-derived macrophages (BMDM) used in our work were obtained following the protocol described by Gomes-Alves and colleagues [16]. In brief, bone marrow (BM) cells were recovered from femurs and tibias of BALB/c mice and differentiated to macrophages in Dulbecco’s modified Eagle’s medium (DMEM; Gibco^®^ Life Technologies Thermo Fisher, Waltham, MA, USA), supplemented with 10% heat-inactivated FBS, 1% minimum essential medium-nonessential amino acids solution (MEM; Gibco^®^ Life Technologies Thermo Fisher, Waltham, MA, USA), 50 U/mL penicillin, 50 μg/mL streptomycin (complete DMEM medium [cDMEM]) and 20% L929 cell-conditioned medium (LCCM) as a source of the macrophage-colony stimulating factor (M-CSF). BM cells distributed into 96-well plates at 3.3 × 10^5^ cells/mL (assuring a final concentration of 5 × 10^4^ cells in each well) and 20% of LCCM and incubated at 37 °C with 5% CO_2_ for 8 days, with cDMEM plus 20% LCCM renewal on the third and sixth days. After these 8 days, the macrophages were incubated with serial dilutions of the compounds for 24 h. The cell viability was then evaluated through the resazurin assay, following the conditions already described, with the percentage of viable macrophages being calculated in relation to control cultures to which only a vehicle (5% DMSO) was added. The half-maximal effective concentrations (EC_50_) were derived from nonlinear regression analysis, using “log(inhibitor) vs. response (three parameters)” function in GraphPad Prism 8 software (GraphPad Software, Inc., La Jolla, CA, USA).

## 3. Results and Discussion

### 3.1. Virtual Screening against AdoMet DC

Considering the metabolic relevance of AdoMet DC to *Leishmania* and the differences between the parasite’s and the human’s AdoMet DC (sequence identity of 27%), this enzyme was selected as the molecular target to use in further computational studies. From the large library of 99 K natural compounds evaluated in the computational study for AdoMet DC, many compounds emerged as potential leads for further development of an antileishmanial drug. By analyzing their complexity, the two tetrahydroacridines most likely to be synthetically obtained were selected (**1** and **2**, Figure 1) and their full synthetic routes were planned, aiming for performing the most efficient and greener procedures.

### 3.2. Chemistry

Considering the compounds indicated in the computational studies (**1** and **2**, Figure 1), the first synthetic step of our work aimed at the preparation of 9-chloro-1,2,3,4-tetrahydroacridines (**5**) in a synthesis accomplished through a POCl_3_-mediated cyclodehydration reaction of 2-amino-4-chlorobenzoic acid (**3**) with cyclohexanone (**4**), a methodology described by Hu et al. (Figure 1) [15]. This reaction was then performed with an excess of POCl_3_ for 2 h at reflux using conventional heating with yields of 41–68%. A microwave (MW)-assisted approach was also performed following specific conditions: potency = 150 W, temperature = 150 °C and time = 30 min, which provided a synthetic alternative with reduced reaction time and better yields (61–70%, Figure 1). The second step consisted of the amination of the freshly synthesized 9-chloro-1,2,3,4-tetrahydroacridines (**5**) with different alkyldiamines in a MW-assisted synthesis (potency = 150 W), using phenol as solvent and NaI as the catalyst, in a reaction that occurred at 150 °C for 30 min, leading to the formation of the desired 9-alkylamino-1,2,3,4-tetrahydroacridines (**6**) with moderate to good yields (31–73%).

Upon obtention of the desired 9-alkylamino-1,2,3,4-tetrahydroacridines (**6**), three different modifications were performed to obtain the compounds suggested by the computational studies, alongside some other structurally similar derivatives. Having compound **1** (Figure 1) as a structural guideline, one of these routes involved the N-acetylation of the 9-alkylamino-1,2,3,4-tetrahydroacridines (**6**) in a MW-assisted reaction (potency = 50 W) with an excess of acetic anhydride, in methanol and using a small amount of sulfuric acid, for 10 min at 100 °C (Figure 1), leading to the formation of the desired N-acetylated products (**7**) in a quantitative manner. Then, another modification consisted of the N-formylation of some 9-alkylamino-1,2,3,4-tetrahydroacridines (**6.e–g; l–n**) through the thermal degradation of DMF, following previous works that described DMF as the source of numerous functional groups, with emphasis on the dimethylamino and formyl groups [17]. Thus, this modification was accomplished through a MW-assisted (potency = 150 W) reaction that occurred in DMF using triethylamine as base, at 120 °C for 30 min, leading to the formation of the desired *N*-formylated 9-alkylamino-1,2,3,4-tetrahydroacridines (**8**) with moderate to good yields (40–88%).

Having compound **2** (Figure 1) as a structural guideline, the dimerization of two tetrahydroacridine fragments linked to a benzoquinone core through a 1,3-diaminopropane chain was performed. This synthesis consisted of a nucleophilic substitution reaction of the 9-alkylamino-1,2,3,4-tetrahydroacridine (**6.b** and **6.i**) to the 2,5-dichloro-1,4-benzoquinone fragment, in a reaction using ethanol as solvent and Na_2_CO_3_ as base (Figure 1) [18]. Due to the high reactivity of both starting materials, the use of an external source of energy was not necessary, being the reaction performed at room temperature for 2–3 h. However, it was observed that for each one of these reactions, two particular derivatives were formed, namely a non-chlorinated (**9.a** [34%] and **9.c** [44%]) and a chlorinated derivative (**9.b** [19%] and **9.c** [29%]), with the formation of the latter being preferred.

It is important to mention that, during the amination of 9-chloro-1,2,3,4-tetrahydroacridines (**5**), some by-products corresponding to the dimerization of the 1,2,3,4-tetrahydroacridines linked through the selected alkyldiamine were also formed (**10**, Figure 2). Even though these compounds were reaction by-products, we still decided to isolate the derivatives (**10**) and evaluate their potential as antileishmanial agents.

Based on the preliminary results that indicated longer alkyl chains as a promising structural feature for the antileishmanial activity of 1,2,3,4-tetrahydroacridine dimers (**10**), we also decided to replace the 1,2,3,4-tetrahydroacridine fragment with a 7-chloroquinoline scaffold. In this case, we followed the same procedure depicted in Figure 1, with 4,7-dichloroquinoline (**11**) reacting with dodecane-1,12-diamine in the presence of NaI as the catalyst and using phenol as solvent (Figure 2). This MW-assisted synthesis was then performed in the following conditions: potency = 150 W; temperature = 210 °C and time = 30 min. However, during this reaction, some monosubstituted derivative was also formed, negatively affecting the yield of the desired dimerization (53%). The entire structural characterization of the synthesized compounds (^1^H, ^13^C and HRMS) is represented in the Appendix A.

### 3.3. Antileishmanial Evaluation against L. infantum Promastigotes

Evaluation of the compounds against *L. (L.) infantum* promastigotes was carried out in steps (Table 1). First, we evaluated the building blocks, namely 9-chloro-1,2,3,4-tetrahydroacridines (**5**), finding that these derivatives showed no activity in the tested range (IC_50_ > 100 μM). Next, we determined the antileishmanial activity of the group of N-acetylated 9-alkylamino-1,2,3,4-tetrahydroacridine (**7**) with different carbon chain linkers between the 1,2,3,4-tetrahydroacridine scaffold and the acetyl group, from two to twelve carbons. The results demonstrated that the length of the chain linker has a crucial influence on the antileishmanial properties of these derivatives (**7**), with considerable levels of activity in derivatives with chain linkers longer than eight carbons. Furthermore, it also became obvious that the presence of a chlorine atom at the 1,2,3,4-tetrahydroacridine fragment (**7.h–n**) might promote increased activities when compared with the simple derivatives (**7.a–g**), but only up to a certain chain length. This is suggested by the fact that, with chain linkers of eight and ten carbons, the chlorinated derivatives presented higher antileishmanial activity, with IC_50_ values of >50 μM (**7.l**) and 5.78 ± 0.65 μM (**7.m**) against >100 μM (**7.e**) and 11.20 ± 0.66 μM (**7.f**). However, with a chain linker of twelve carbons, this effect appears to be inverted, leading to a higher activity of the unsubstituted 1,2,3,4-tetrahydroacridine derivative (**7.g**, IC_50_ = 1.86 ± 0.48 μM) in contrast with the chlorinated derivative (**7.n**, IC_50_ = 4.47 ± 0.79 μM).

The *N*-formylated 9-alkylamino-1,2,3,4-tetrahydroacridines (**8**) were also evaluated for their antileishmanial properties. Similarly to the *N*-acetylated derivatives (**7**), the antileishmanial activity of this group (**8**) increased with the length of the carbon chain linker length. Interestingly, however, this second group of compounds (**8**) presented higher antileishmanial activities than those observed with *N*-acetylated derivatives (**7**), suggesting that the introduction of bigger terminal groups might negatively influence the compound’s antileishmanial activities. Furthermore, it was possible to verify that, for derivatives with carbon chain linkers up to ten carbons, the presence of a chloride atom in the 1,2,3,4-tetrahydroacridine scaffold promoted higher antileishmanial activities (**8.b**, IC_50_ = 5.64 ± 0.65 μM compared with **8.e**, IC_50_ = 3.23 ± 0.41 μM). However, when it comes to derivatives containing a twelve-carbon chain linker, this effect seems to be minimized, with both derivatives **8.c** (IC_50_ = 1.69 ± 0.22 μM) and **8.f** (IC_50_ = 1.55 ± 0.38 μM) showing similar antileishmanial activity levels.

Finally, we evaluated the series of dimers obtained throughout the entire project, including derivatives with simple alkyldiamine linkers (**10**, Figure 2), with alkylated benzoquinone linkers (**9**, Figure 1), and also a quinoline dimer containing a twelve-carbon chain linker (**12**, Figure 2). Considering the first group of compounds from this series, the results clearly demonstrated that the dimerization of this type of scaffold (**10**) leads to more active antileishmanial derivatives, with particular focus on those containing a twelve-carbon chain linker (**10.g**, IC_50_ = 0.37 ± 0.06 μM and **10.m**, IC_50_ = 1.92 ± 0.21 μM, respectively). Additionally, in these derivatives (**10**), the relation between the length of the carbon linker and the compound’s antileishmanial properties can be clearly verified, since longer chain linkers promote higher antileishmanial activities. When it comes to the dimers containing a benzoquinone fragment in the linker (**9**), only one derivative (**9.d**) demonstrated promising antileishmanial activity with an IC_50_ value of 2.17 ± 0.45 μM. Of note, the quinoline dimer (**12**) revealed to be one of the most active antileishmanial compounds from our entire work (IC_50_ = 0.60 ± 0.11 μM), proving that the replacement of 1,2,3,4-tetrahydroacridine scaffold for a 7-chloroquinoline core might be a promising approach for lead optimization.

### 3.4. Toxicity against Mammalian Macrophages

Upon testing the antileishmanial activity of the compounds referred to previously, the most promising derivatives were evaluated towards their toxicity against macrophages. This evaluation intended to assess compound safety and is a crucial step in any antileishmanial drug development project. The results demonstrated that, except for compound **12**, the entire series of the evaluated compounds presented high levels of toxicity, with the majority of them killing at least 90% of the macrophages at the concentration of 10 μM (Table 1). As a result, even though this series of 1,2,3,4-tetrahydroacridines showed considerable antileishmanial properties, no subsequent studies will be carried out. Importantly, however, the replacement of the 1,2,3,4-tetrahydroacridine scaffold by 7-chloroquinoline resulted in a significant improvement in both compounds’ antileishmanial activity and toxicity levels, culminating in a compound with a therapeutic index of 19.48. Thus, these results suggest that, by replacing the 1,2,3,4-tetrahydroacridine scaffold with a 7-chloroquinoline, we might be able to develop a series of active and safer antileishmanial compounds using the remaining structural features indicated throughout this work.

### 3.5. Molecular Docking of the Most Promising Antileishmanial Agents

Some of the most promising antileishmanial derivatives from this series (Table 1) were submitted to molecular docking calculations in the enzyme’s active site. Once again, the influence of the structural variations in the binding mode of our compounds to the enzyme were visually inspected using the PyMol 2.5.4 software. The results obtained through these final molecular docking studies were intended to give us some insight about the way distinct substitution patterns influence the compound–enzyme interactions. For instance, the biological evaluation of this series of 1,2,3,4-tetrahydroacridine derivatives demonstrated that, in general, the presence of a chlorine atom at C-6 of the 1,2,3,4-tetrahydroacridine scaffold promotes higher antileishmanial activity levels. In terms of the *N*-acetylated 9-alkylamino-1,2,3,4-tetrahydroacridines (**7**), the existence of this chlorine atom does not affect the close interactions between the amino groups of the derivatives and the enzyme’s THR-98 residue (Figure 3). However, this substitution pattern promotes the formation of a novel hydrogen bond between the chlorine atom and the enzyme’s ILE-278 residue, since this chlorine atom can also act as a weak hydrogen bond acceptor [19], leading to more stable compound–enzyme complexes. Furthermore, the higher antileishmanial properties of derivatives containing longer carbon chains might be explained by the close interaction of the carbonyl group with the enzyme’s SER-229 residue. Finally, it is also possible to verify that this type of scaffold is stabilized by one long π–π interaction between the 1,2,3,4-tetrahydroacridine nucleus and the enzyme’s PHE-42 residue.

Considering the 1,2,3,4-tetrahydroacridine dimers (**10**), the same type of compound–enzyme interactions is present, being mostly characterized by hydrogen bonds to the enzyme’s THR-98 and, in the case of the chlorinated derivatives, ILE-278 residues (Figure 4). Nevertheless, some peculiarities verified in the docking poses of these derivatives might explain the distinct antileishmanial properties of this type of compound (**10**). For instance, for derivatives with short chain linkers, the compound–enzyme complexes seem to be stabilized by the existence of two long π–π interactions with the enzyme’s PHE-42 and PHE-227 residues (Figure 4A,C). In turn, for derivatives with long chain linkers (Figure 4B,D), the interactions of the compound–enzyme complexes have one considerable difference depending on the substitution pattern of the 1,2,3,4-tetrahydroacridine scaffold. This difference is mainly characterized by the loss of a π–π interaction between the second 1,2,3,4-tetrahydroacridine scaffold of the chlorinated derivative and the enzyme’s PHE-227 residue, while the unsubstituted derivative maintains this same interaction. This evidence might explain the higher antileishmanial properties verified by the unsubstituted derivatives with chain linkers longer than ten carbons when compared with their chlorinated homologous. Considering the two final 1,2,3,4-tetrahydroacridine dimers evaluated (**9**), the same type of interactions already described are verified, consisting of hydrogen bonds with the enzyme’s THR-98 and ILE-278 (in the case of the chlorinated derivative) residues, and π–π interactions with the enzyme’s PHE-42 and PHE-227 residues (Figure 4E,F). However, these interactions are not enough to clearly explain the considerable difference of antileishmanial activity verified between these two similar derivatives, which might suggest the existence of a different molecular target being affected by this type of derivative (**9**).

Finally, one of the most active compounds reported here was synthesized to assess the influence of other scaffolds linked to the terminal amino groups of the alkyldiamines, replacing the 1,2,3,4-tetrahydroacridine scaffold with a 7-chloroquinoline core (**12**). However, it was interesting to verify that this compound interacts with neither the catalytic nor the allosteric sites of our target enzyme, which might suggest that some other target may be responsible for the antileishmanial activity of this compound.

## 4. Comments and Conclusions

Considering the evaluation of the antileishmanial properties of this series of compounds, 1,2,3,4-tetrahydroacridine derivatives, a total of forty-one compounds were evaluated against *L. (L.) infantum* promastigotes. Throughout this entire series of 1,2,3,4-tetrahydroacridine derivatives, three major structural features were studied: (i) the influence of the presence/absence of a 6-chlorine atom on the 1,2,3,4-tetrahydroacridine scaffold; (ii) the effect promoted by the functional group or scaffold introduced in the terminal amino group of the alkyl chain; and (iii) the effect of different chain lengths on the antileishmanial properties of these compounds. The results clearly showed that the length of the chain linker has a crucial effect on the compounds’ antileishmanial properties, with longer chain linkers (8–12 carbons) originating highly active derivatives. Furthermore, the presence of a 6-chlorine atom at the 1,2,3,4-tetrahydroacridine scaffold also promoted a general increase in activity. This effect is also dependent on the linker length. Finally, the functional group/scaffold introduced into the terminal amino group of the alkyl chain also showed to have some influence in the compounds’ antileishmanial properties, with a second 1,2,3,4-tetrahydroacridine scaffold originating the most active group (**10**) of this series of compounds, followed by the *N*-formylated (**8**) and *N*-acetylated (**7**) derivatives (Figure 5).

Unfortunately, despite evident antileishmanial properties, these derivatives are not suitable for further studies towards the development of novel antileishmanial agents since they present high levels of toxicity towards macrophages. To overcome this limitation, one final derivative was synthesized, where the 1,2,3,4-tetrahydroacridine scaffold was substituted by 7-chloroquinoline in a dimer containing a twelve-carbon length linker (**12**). Remarkably, this approach led to a derivative that maintained high antileishmanial levels (IC_50_ = 0.60 ± 0.11 μM) with considerably lower levels of toxicity (EC_50_ = 11.69 ± 3.96 μM, TI = 19.48). This final result proves that, despite the toxicity presented by the 1,2,3,4-tetrahydroacridine derivatives, the structural information gathered throughout this project might guide future works for the development of active and safe antileishmanial agents.

## Data Availability

Not applicable.

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
