# Peer review of "Structure–Activity Relationship Studies of 9-Alkylamino-1,2,3,4-tetrahydroacridines against Leishmania (Leishmania) infantum Promastigotes"

_pharmaceutics, 2023, doi:10.3390/pharmaceutics15020669_

Round 1

Reviewer 1 Report

Accepted after  minor correction as at the manuscript at the attachment

Reviewer 2 Report

The authors of the article describe the synthesis of 9-alkylamino-1,2,3,4-tetrahydroacridines derivatives and their activity against L. infantum promastigotes.

The introduction is written correctly, but the synthetic chemical part arouses my attention because it is written unclearly, the regulations differ from the standard presentation of chemical regulations, and they leave many inaccuracies. The docking tests are correct. A bit too long summary section.

Main remarks:

1) For all equipment or key reagents used, please provide the manufacturer, country and city.

2) Line 116-117: "S" should be in italics, the rest "adenosylmethionine decarboxylase" in normal type

3) Line 96: "thieves"? please check and correct.

4) Section 2.3 - Synthesis:

- what was the volume of the teflon tube? please add in each synthesis.

- line 192: "and of phenol" - please check and correct.

- lines 196-197 and 317-318: what volume of 10% KOH was used? please add.

- line 236: what sulfuric acid was used? please add.

- lines 238-239: between what and what was the extraction conducted? solvent is missing.

- line 251: "amber"? shouldn not it be amber?

- line 266: 265-266: between what and what was the extraction conducted? solvent is missing. how much ethyl acetate was used?

5) Please add the numbering of the atoms in the structures.

6) Section 3.3. - no reference compound in bioassays. Why? This would allow us to immediately check whether the obtained derivatives are better than commercially used compounds or not.

7) Supplementary info: I see no point in repeating the recipes and spectroscopic data from the main text of the manuscript in SI. Instead, please put all 1H and 13C NMR spectra and HRESI-MS spectra into the SI.

Reviewer 3 Report

The manuscript “SAR studies of 9-alkylamino-1,2,3,4-tetrahydroacridines against L. infantum promastigotes” by Silva et al describes the results of computational analysis focusing on a specific enzyme of the parasite, S-adenosylmethionine decarboxylase, with several 1,2,3,4-tetrahydro-acridines. Following are my comments and suggestions:

- Although written well, the manuscript is too long for its findings. It should be more concise to about 60% or less of the current length, focusing on the theme/title without redundantly citing information already reported.

- Title (suggested): “Structure-activity relationship studies of 9-alkylamino-1,2,3,4- tetrahydroacridines against Leishmania (L.) infantum promastigotes”

- The Introduction sections need to be reduced to two-thirds of the current length, focusing on the theme/title.

- Materials and Methods section should be written more concisely, avoiding repetition of methods/procedures already-reported.

- The Conclusion section is too long. Is it replaced by Discussion section ? ; Discussion section is not available.

- The use of amastigote stage of the Leishmania parasite should be considered in future study. 

Reviewer 4 Report

The author's research is very meaningful, 1,2,3,4-testrahydroidines is expected to be able to treat Leishmaniasisis as novel potential antileishman agents. There is no major problem with the overall design of the article, but there are some details to be revised. Thus, I suggest that the article can be published after a minor revision.

The specific content is as follows,

1.      General is confusing as a 2.1 section title, and it is recommended to modify it into specific content.

2.      Line 99, 110, et al, For example, thin layer chromatography (TLC)  and Buchi Melting Point B-540 equipment should give the producer name, the place of production and the country's information. Please check and revise the full text.

3.      The authors used HRMS-ESI, but did not give information about the instrument and conditions of use. Please add information to the article.

4.      Figures 3 and 4 are recommended to re -typeset to make the pictures beautiful.

5.      3. Results and 5. Conclusions, is the article missing Section 4. Discussion section? In addition, the conclusion of the article is too long, including figure 4. The conclusion is not recommended to have Figure, in addition, the order of Figure is wrong. It is recommended that this part of the discussion section and simplify the conclusion. Article structure titles, including small titles, need to be pondered over and over again, be more specific.

Reviewer 5 Report

Authors have written a research article titled “SAR studies of 9-alkylamino-1,2,3,4-tetrahydroacridines against L. infantum promastigotes”. The manuscript can be considered for publication after major revision with consideration following points.

1.      In the abstract “chemotherapy is problematic” can be corrected as “chemotherapy, which is problematic”.

2.      At the end of the introduction section, authors should mention the aim of the articles.

3.      This article may confuse the readers, whether it is a review or research article since most of the sections contain only general procedure.

4.      Synthesis processes are not clear for understanding.

5.      Authors have presented many synthesis procedures but none of them were supported with spectral graphs such as NMR or Mass etc.

6.      Additionally, the conclusion should be based on the outcome of the overall study, devoid of references or images.

Round 2

Reviewer 3 Report

The manuscript is revised appropriately responding to this reviewer’s comments; besides, the following comments and suggestions would be better to be further considered:

- Title: replace "Leishmania (L.) infantum" by Leishmania (Leishmania) infantum  

Abstract

- Line 23, replace “Leishmania infantum” by Leishmania (Leishmania) infantum

1.Introduction section

- Although improved extensively, still however, the section might be further concise, focusing on the theme/aim/title.

- Usually, inserting “Figure” is not recommendable in the Introduction section; so, “Figure 1” might be deleted; if necessary, mention it in the “Results and Discussion section” appropriately.

-Line 70, replace “T. brucei” by “Trypanosoma brucei”

-Lines 78-80; Move to the end of the Introduction section as the last phrase (aim of the study should be mentioned and/or insisted clearly); then, I would like to recommend some modification as follows:

“The aim of this study is to find potential hit compounds against L. (Leishmania) infantum, based on the virtual screening of a considerably large chemical library of natural compounds, belonging to a wide range of families of compounds, against AdoMet DC.”

-Line 81: replace “Within this library” by “Within the library”

-Line 84, delete “Throughout the years”

-Line 91, replace “ leaving us some room to develop” by “leaving some possibilities to develop”

 2.Materials and Methods

-Line 166, replace “L. infantum” by “L. (L.) infantum”

-Line 352, replace “Leishmania infantum” by L. (L.) infantum

3.Results and Discussion

- Line 476, replace” L. infantum” by L.(L.) infantum

- Line 604; I recommend inclusion of “Comments” as follows: 4. Comments and Conclusion

Author Response

Dear Reviewer 3,

Once again, we sincerely appreciate the interest demonstrated in our manuscript and all the insightful comments and suggestions made to improve our final version. All the suggestions made were accepted and promptly followed in order to meet both reviewer's and journal's requirements.

Yours sincerely,

Carlos Silva.

Reviewer 5 Report

Can be accepted for publication 

Author Response

Dear Reviewer 5,

Once again, we deeply appreciate the interest demonstrated in our manuscript and all the insightful comments and suggestions made to improve our final version.

Yours sincerely,

Carlos Silva.